# Life habits and evolutionary biology of new two-winged long-proboscid scorpionflies from mid-Cretaceous Myanmar amber

Xiaodan Lin[1,2], Conrad C. Labandeira[1,2,3], Chungkun Shih[1,2], Carol L. Hotton[2,4] & Dong Ren [1]

Long-proboscid scorpionflies are enigmatic, mid-Mesozoic insects associated with gymnosperm pollination. One major lineage, Aneuretopsychina, consists of four families plus two haustellate clades, Diptera and Siphonaptera. One clade, Pseudopolycentropodidae, from mid-Cretaceous Myanmar amber, contains *Parapolycentropus*. Here, we newly establish *Dualula*, assigned to Dualulidae, constituting the fifth lineage. *Parapolycentropus* and *Dualula* lineages are small, two-winged, with unique siphonate mouthparts for imbibing pollination drops. A cibarial pump provides siphonal food inflow; in *Dualula*, the siphon base surrounds a hypopharynx housing a small, valved pump constricted to a narrow salivary duct supplying outgoing enzymes for food fluidization. Indirect evidence links long-proboscid mouthpart structure with contemporaneous tubulate ovulate organs. Direct evidence of gymnospermous *Cycadopites* pollen is associated with one *Parapolycentropus* specimen. *Parapolycentropus* and *Dualula* exhibit hind-wing reduction that would precede haltere formation, likely caused by *Ultrabithorax*. Distinctive, male Aneuretopsychina genitalia are evident from specimens in copulo, supplemented by mixed-sex individuals of likely male mating swarms.

[1] College of Life Sciences, Capital Normal University, 100048 Beijing, China. [2] Department of Paleobiology, National Museum of Natural History, Smithsonian Institution, Washington, DC 20013, USA. [3] Department of Entomology, University of Maryland, College Park, MD 20742, USA. [4] National Center for Biotechnology Information, National Library of Medicine, Bethesda, MD 20892, USA. Correspondence and requests for materials should be addressed to C.C.L. (email: labandec@si.edu) or to D.R. (email: rendong@mail.cnu.edu.cn)

Long-proboscid scorpionflies (Mecoptera) have a long evolutionary history of interacting with plants in Eurasia beginning during the late Permian and ending in the mid Cretaceous. The earliest scorpionfly lineage with long-proboscid mouthparts, defined by a projecting siphon, was Nedubroviidae from late Permian Russia[1,2]. Taxa within this small-bodied lineage survived the Permian–Triassic ecological crisis into the ensuing Triassic[2,3], supplemented by two lineages of long-proboscid scorpionflies, Mesopsychidae and Pseudopolycentropodidae. During the mid Mesozoic, with proliferation of additional taxa from three other insect orders, there minimally were 13 independent originations of long-proboscid mouthparts[4]. These long-proboscid groups included Mecoptera[5–7] (scorpionflies, three originations from this study), Neuroptera[8,9] (lacewings, three originations), Lepidoptera[10,11] (moths and butterflies, one origination), and Diptera[12,13] (true flies, six originations).

For Mecoptera, all long proboscid taxa historically were contained within the presumably monophyletic, latest Paleozoic to mid-Mesozoic lineage, Aneuretopsychina[14], which comprised four families—Nedubroviidae[2], Mesopsychidae[3], Pseudopolycentropodidae[15–17], and Aneuretopsychidae[14]. The Nedubroviidae consisted of *Nedubrovidia shcherbakovi* and three other congeneric species from Late Permian European Russia at ca. 254 Ma[2]. The last lineage, Pseudopolycentropodidae, is documented from several mid-Mesozoic deposits, including *Parapolycentropus burmiticus* and *P. paraburmiticus*[16,17] from mid-Cretaceous Myanmar (Burmese) amber at 99 Ma[18]. Aneuretopsychina had two intervals of diversification—an earlier, modest resurgence during the latter Triassic, and a greater speciation interval from the Middle Jurassic to Early Cretaceous. Understanding of Aneuretopsychina biology has increased greatly from examination of compression deposit occurrences[2,3,6,7,19,20], to a recent focus on late appearing lineages from Myanmar amber[16,17,21]. Examination of these amber taxa soon before extinction of Aneuretopsychina now can provide more finely resolved details of the life habits and evolutionary biology of this bizarre[17] group of insects.

In this contribution, we provide long-proboscid scorpionfly data from Middle Jurassic compression deposits of Northeastern China, and mid-Cretaceous amber from Northern Myanmar. These taxa are placed into a phylogenetic context within lineages of extant and extinct Mecoptera that are linked phylogenetically to Amphiesmenoptera and Neuroptera outgroups, as well as ingroups including seven basal lineages of Diptera and Siphonaptera. The new family that we establish constitutes a modification of siphonate, pseudopolycentropodid-type mouthparts not documented in any other, known, long-proboscid group. A sufficiently well preserved number of insect specimens have been marshalled to provide evidence for documenting transformation of the mecopteran hind wing into a haltere-like structure by reference to the *Ultrabithorax* homeotic gene system in *Drosophila*. We document structurally well-preserved male genitalia from compression Mesopsychidae and Pseudopolycentropodidae that are compared to the amber taxa, the latter including an in copulo pair, revealing stereotyped patterns of scorpionfly genitalia structure during the mid Mesozoic. Similarly, from three amber pieces containing mixed sex and species congregations of two *Parapolycentropus* species, we describe evidence for lekking swarms. Our documentation of a new Pseudopolycentropodidae lineage establishes a new, long-proboscid family that, together with its closely related sister-taxon, possesses a unique, new mouthpart type that allows comparisons to other extinct and extant long-proboscid morphologies. From a variety of indirect and direct evidence, we provide an explicit explanation of the feeding mechanisms of these taxa, and their

association with gymnosperm hosts based on mouthpart structure, host-plant ovulate organ morphology, and adjacent pollen. Our multifaceted study should enlarge knowledge of long-proboscid scorpionfly ecology and their life habits from the deep past.

## Results

**Phylogenetic analysis.** We conducted a phylogenetic analysis to clarify the taxonomic position of new family and understand the relationships of long-proboscid clade (Aneuretopsychina), as well as Mecoptera in general. This analysis included two relevant representatives of Neuroptera and Amphiesmenoptera as outgroups, and Siphonaptera and seven early appearing, basal lineages of Diptera as ingroups. The input data consisted of 27 major lineages of extinct and extant Mecoptera that sampled a wide diversity of body form (Supplementary Data 1 and 2). A full complement of 51 morphologic characters coding 37 total taxa represented head, wing, leg, thorax and abdominal features (Supplementary Note 1).

A maximum parsimony analysis yielded 93 most parsimonious trees. The strict consensus result (Fig. 1a) has a tree length of 159 steps, consistency index (CI) of 0.34 and retention index (RI) of 0.73. Morphological characters were optimized with parsimony on all most-parsimonious trees, showing only unambiguous changes. We chose the twenty-fourth generated tree (Fig. 1b) as the most suitable tree based on a summary of pre-existing phylogenetic conclusions from several previous analyses of Panorpoidea sensu stricto, including Dinopanorpidae, Orthophlebiidae, Panorpidae, Panorpodidae[22] and basal Diptera[23] (Supplementary Fig. 1). Bootstrap values are shown in Fig. 1.

The phylogenetic analysis provided five important results (Fig. 1). First, Mecoptera are a paraphyletic group and Thaumatomeropidae and Kaltanidae are basalmost taxa. Second, Meropeidae and Eomeropidae are more basal than other extant and extinct families of Mecoptera. Third, long-proboscid Aneuretopsychina are a paraphyletic group, with *Parapolycentropus* + Dualulidae having closer affinities to basal Diptera + Siphonaptera. However, the unknown mouthpart structure of Liassophilidae and Permotanyderidae calls into question the phylogenetic status of Aneuretopsychina, requiring further investigation with additional, well-preserved fossils from these lineages that reveal mouthpart structure. Fourth, Aneuretopsychina, including Pseudopolycentropodidae, Aneuretopsychidae, Mesopsychidae, and Nedubroviidae, and possible long-proboscid Liassophilidae and Permotanyderidae, are a sister-clade to a (*Parapolycentropus* + Dualulidae) + (basal Diptera + Siphonaptera) clade. Fifth, Aneuretopsychina likely were phenetically similar to the immediate ancestor of Diptera, particularly as the *Parapolycentropus* + Dualulidae lineage exhibits close affinities to basal dipteran taxa and Siphonaptera.

**Systematic palaeontology.**

<div align="center">

Order Mecoptera Packard, 1886
Suborder Aneuretopsychina Rasnitsyn and Kozlov, 1990.
Family Dualulidae Lin, Shih, Labandeira and Ren, fam. nov.

</div>

**Type genus**. *Dualula* Lin, Shih, Labandeira and Ren gen. nov. (Figs. 2 and 3; head and mouthparts reconstructed in Fig. 4).
**Diagnosis**. Body size small, length ca. 7.6–8.4 mm (excluding antennae and proboscis). Head triangular in dorsal view, with long, narrow proboscis. Antennae filiform, slender; shorter than proboscis. Compound eyes large,

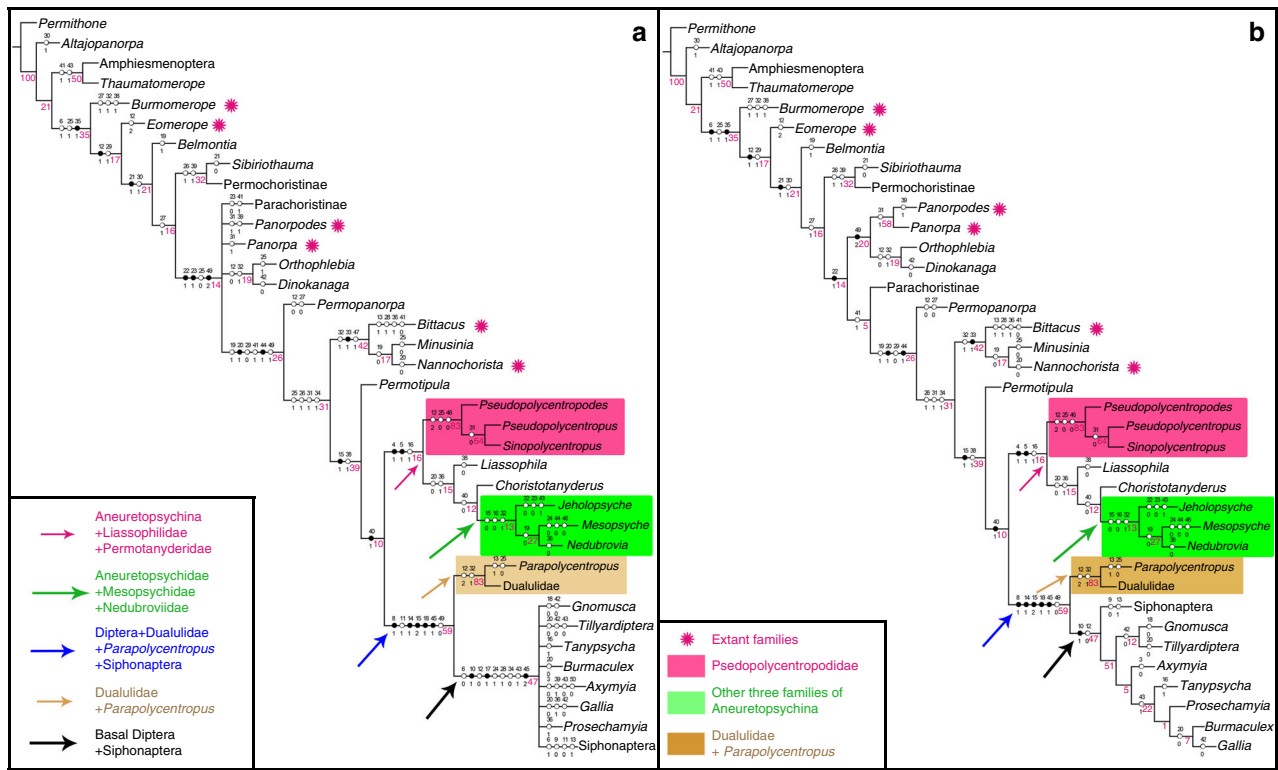

**Fig. 1** Results of the phylogenetic analysis by NONA. **a** Strict consensus tree of 93 maximum parsimony trees from NONA. **b** The twenty-fourth maximum parsimony tree from the NONA analysis. Open circles are plesiomorphic characters; solid black circles are apomorphic characters. The numbers above branches are characters; numbers below the branches are character states; and magenta numbers below the branches are bootstrap values in **a** and **b**. Colored arrows refer to the following clades or lineages: magenta = Aneuretopsychina sensu lato; green = Aneuretopsychidae + (Mesopsychidae + Nedubroviidae); blue = (*Parapolycentropus* + Dualulidae) + (basal Diptera + Siphonaptera); brown = *Parapolycentropus* + Dualulidae; black = basal Diptera and Siphonaptera

separated. Prothorax and metathorax small, mesothorax comparatively enlarged. Legs slender; two claws at the end of pretarsus. Forewing long, ovoidal, slightly rounded apex. Sc relatively short, extending to C near Rs bifurcation, with an anterior branch slightly distal to or at same level of R bifurcation. $R_1$ single and extending much beyond $Rs_{1+2}$ forking; Rs with four branches, $Rs_{1+2}$ forking considerably distal to $Rs_{3+4}$; Rs forking proximal to M. M with four branches, $M_{1+2}$ bifurcation considerably distal to $M_{3+4}$; Rs originating from $R_1$ distinctly distal to M from CuA. CuA and CuP single; stem of M curved with an almost right angle; anal area relatively narrow, two or three anal veins present; two crossveins between CuA and CuP; one a1-a2 present. Thick, short setae on the membrane from R forking to tip of wing; several long setae on entire margin. Hind wing degraded to a minute, tubular-shaped lobe. Female abdomen with 11 segments, but male only nine visible segments. Female cercus with two segments; male ameristic. Male claspers very robust, with bent dististylus. **Included genus**. Type genus only.

Genus *Dualula* Lin, Shih, Labandeira and Ren, gen. nov.

**Type species**. *Dualula kachinensis* Lin, Shih, Labandeira and Ren, sp. nov. (Figs. 1–3, Supplementary Figs. 2–5). **Etymology**. The generic name refers to a combination of *duo-ae* (Latin, meaning 'two' or 'dual') and *–alula-ae* (Latin, a diminutive variant of wing, *ala-ae*, meaning 'a tiny wing' or 'a small appendage'). This designation refers to

the highly miniaturized hind wings of this genus. The gender is female.
**Diagnosis**. As for the family by monotypy.

*Dualula kachinensis* Lin, Shih, Labandeira and Ren, sp. nov.

**Etymology**. The specific epithet is derived from the northern state of Kachin in Myanmar, where the first discovered species of *Dualula* was found, Latinized to *Kachin –ensis*.
**Diagnosis**. As for the genus by monotypy.
**Holotype**. See Figs. 2 and 3; specimen CNU-MEC-MA-2014001. A female with partially preserved body and forewings; complete proboscis and hind wings: Left forewing length at least 9.84 mm, width 2.63 mm; right forewing length 9.68 mm, and width 2.55 mm. Body length 8.12 mm (excluding proboscis and antennae). Proboscis length 3.23 mm; right antenna length at least 2.23 mm.
**Paratypes**. See Supplementary Figs. 2–5, specimens CNU-MEC-MA-2017016 and CNU-MEC-MA-2017017. Female with almost completely preserved body and wings; CNU-MEC-MA-2017016: Right forewing length 9.27 mm, width 2.30 mm; left forewing length 8.78 mm, and width 2.25 mm. Body length 8.4 mm (excluding proboscis and antennae). Proboscis length (as preserved) 2.72 mm; right antenna length 2.13 mm.

Male with completely preserved body and wings, CNU-MEC-MA-2017017: Right forewing length 7.35 mm, width 1.87; left forewing length 6.85 mm, width 1.91 mm. Body

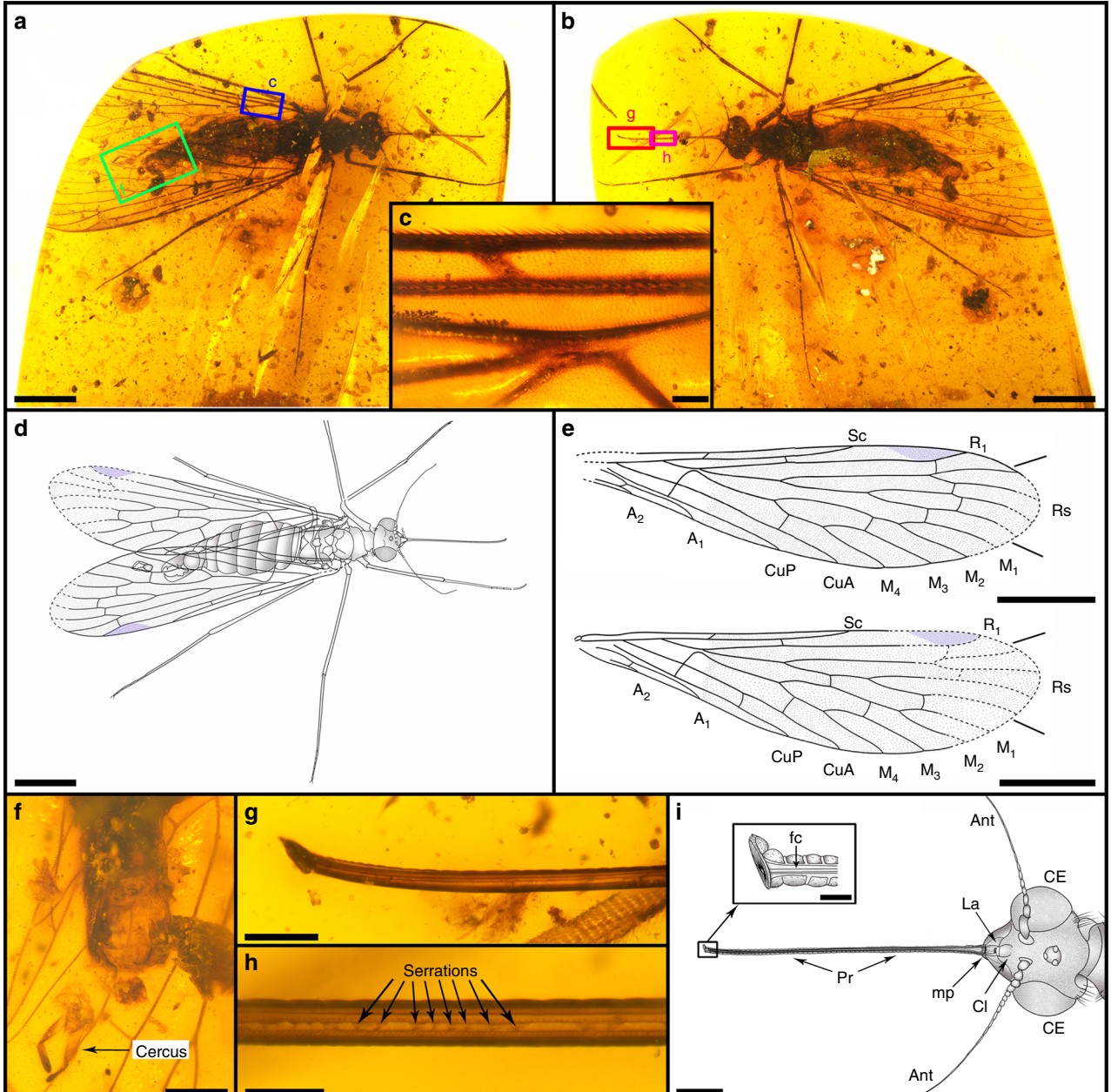

**Fig. 2** Photos and line drawings of holotype with details of the proboscis and genitalia. *Dualula kachinensis* gen. et sp. nov., CNU-MEC-MA-2014001, female. **a** Holotype in dorsal view. **b** Holotype in ventral view. **c** Details of setae on the margin and membrane of the left forewing, enlarged from blue template in **a**. **d** Overlay drawing of holotype in dorsal view. **e** Line drawing of forewings. Above is the right forewing and below is the left forewing. **f** Female genitalia in ventral view, enlarged from green template in **a**. **g** Proboscis terminus, with details in ventral view, enlarged from the larger magenta template in **b**. **h** Proboscis midsection in ventral view, enlarged from the smaller magenta template in **b**. **i** Overlay drawings of the head and proboscis tip details, based on **a**. Ant antenna, CE compound eye, Cl clypeus, fc food canal, La labrum, mp maxillary palp, and Pr proboscis. Scale bars represent 2 mm in **a**, **b**, **d** and **e**; 0.5 mm in **f** and **i**; 0.2 mm in **g**; 0.1 mm in **c** and **h**; and 0.05 mm in proboscis tip from **i**

length 7.62 mm (excluding proboscis and antennae). Proboscis length 1.82 mm; left antenna length 2.08 mm.
**Horizon and Locality**. Hukawng Village, Kachin State, northern Myanmar; Upper Cretaceous (earliest Cenomanian), 98.79 ± 0.62 Ma[18].

**Description**. See Figs. 2–4; Supplementary Note 1; Supplementary Figs. 2–5; Supplementary Data 4.
Further systematic paleontological details of the newly described taxon, *Dualula kachinensis* gen. et sp. nov., related *Parapolycentropus*, and associated issues are provided in Supplementary Note 4. Linked figures showing morphological

features of the mouthparts, head, thorax, wings, legs, abdomen, and genitalia are given in Figs. 1–3 and Supplementary Figs. 2–5. The preservational status and geological provenance of 13 Pseudopolycentropodidae species are given in Supplementary Table 1.

**The long-proboscid condition in Aneuretopsychina.** Currently, there are four commonly occurring groups of long-proboscid, siphonate insects[24]. The first group consists of moths and butterflies (Glossata), a major defining feature is the siphonate proboscis[25]. The second group consists of about eight, major

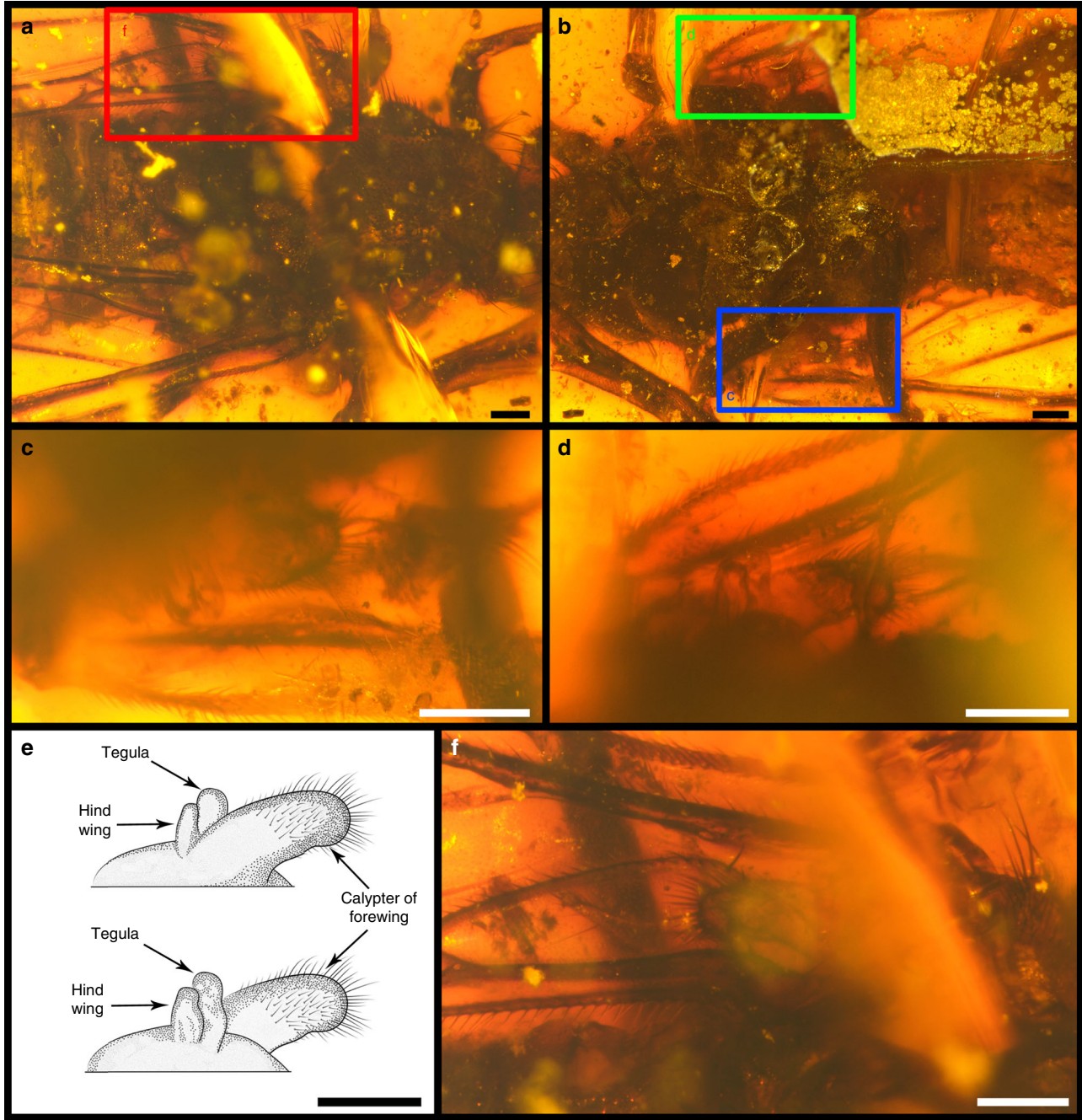

**Fig. 3** Hind wings of holotype *Dualula kachinensis* gen. et sp. nov. of CNU-MEC-MA-2014001. **a** Thorax in dorsal view. **b** Thorax in ventral view. **c** Right hind wing in ventral view, enlarged from blue template in **b**. **d** Left hind wing in ventral view, enlarged from green template in **b**. **e** Line drawings of left hind wing, above in dorsal view and below in ventral view. **f** Left hind calypter in dorsal view, enlarged from the red template in **a**. Scale bars represent 0.2 mm in **a**–**f**

family-level fly lineages in the Brachycera, most of which convergently evolved similar long-proboscid mouthparts, although differences exist in overall form, aspect ratio and surface ornamentation[26,27]. Two other groups of modern, long-proboscid insects are Coleoptera (beetles) and Trichoptera (caddisflies) that occasionally evolved the long-proboscid condition in nonspeciose lineages[28,29]. Modern bearers of long-proboscid mouthparts are similar to the spectrum of groups that possessed long-proboscid mouthparts from the Middle Jurassic to mid-Cretaceous (170 to 95 Ma)[1]. The fourth, long-proboscid, mid-Mesozoic groups consisted of major extinct lineages within Mecoptera[5,6,17] and Neuroptera[8,9,21]. By contrast, mid-Mesozoic long-proboscid

lineages of Diptera[12,13] and Lepidoptera[10,11] overwhelmingly are extant. It is notable that of the four mid-Mesozoic groups, Mecoptera and Neuroptera no longer have long-proboscid forms, and often are biogeographical relicts[9] (Supplementary Note 2).

Within Mecoptera, the historically defined Aneuretopsychina[14] contains four, major, long-proboscid lineages: Nedubroviidae[2], Mesopsychidae[30], Aneuretopsychidae[11] and Pseudopolycentropodidae[20]. The Nedubroviidae[2] are an obscure late Permian to Middle Triassic group for which few details of the head and mouthparts are known, other than a prominent triangular labrum and an incomplete 0.32 mm long proboscis with a food canal that

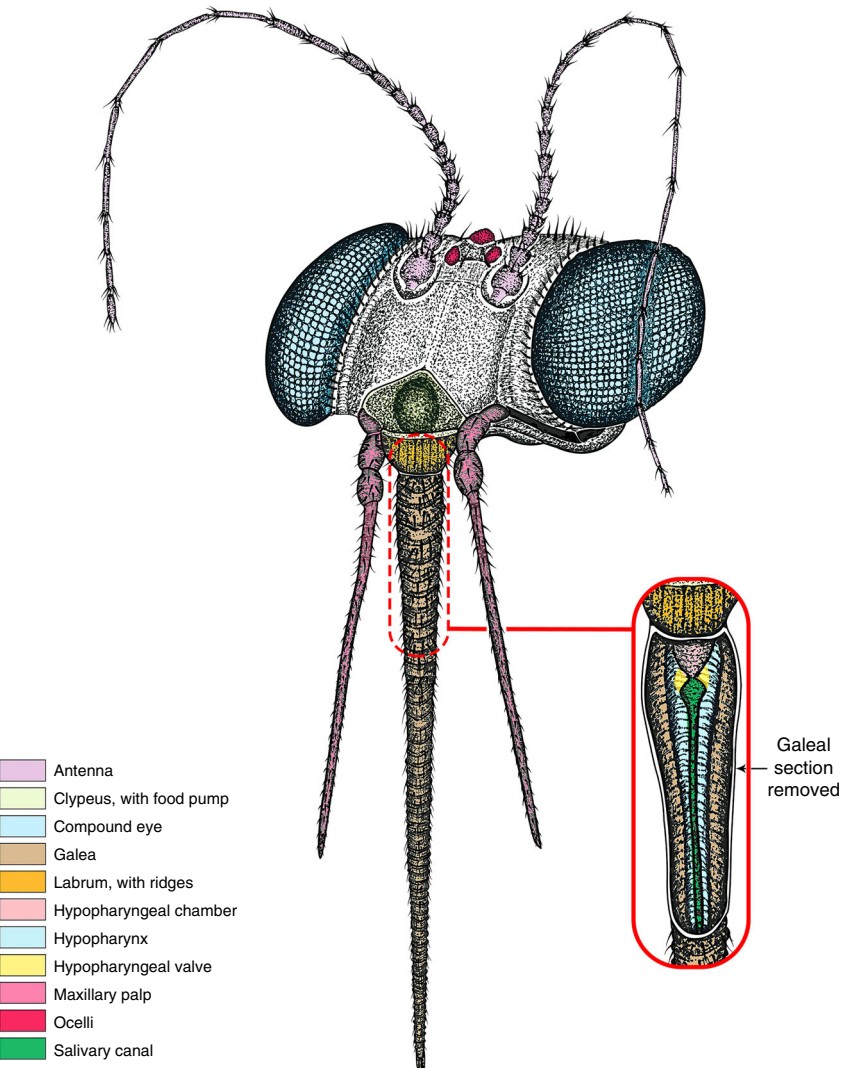

**Fig. 4** Reconstruction of head and mouthparts of male *Dualula kachinensis* gen. et sp. nov. The drawing is based mostly on specimen CNU-MEC-MA-2017017, supplemented by CNU-MEC-MA-2014001 and CNU-MEC-MA-2017016. A section representing the upper proximal third of the galeae has been removed to reveal features below of the pharyngeal pump. This subfigure was created from microscope photographs of the amber specimen by Conrad Labandeira as a hand drawing modified in Adobe Photoshop CC by Xiaodan Lin. Scale bar represents 0.5 mm

is missing a terminus. Better preserved Mesopsychidae are a late Permian to Early Cretaceous lineage bearing a long, forwardly directed (prognathous), siphonate proboscis constructed of maxillary galeae interlocked by a tongue-and-groove suture and housing a central food tube[6,30]. The proboscis terminus houses two, laterally placed ovoidal pseudolabellae and an up to a 11.2 mm long proboscis shaft that has an external surface covered, sometimes sparsely, by randomly positioned, thick setae, but lacking other ornamentation such as transverse ridges. At the proboscis base are laterally positioned, adpressed, three-articled maxillary palps and a domed clypeus suggesting a cibarial pump. Direct evidence for a hypopharynx (pharyngeal pump and salivary duct) is lacking, although functional considerations and the presence of a third proboscis element between separated galeae in one specimen indicates its presence[6].

The known proboscis of Aneuretopsychidae is 8.5 mm long and directed rearward (opisthognathous), a feature differing from all other Aneuretopsychina[6,14,19]. The proboscis has an outer surface of transverse, annular ridges bearing perpendicularly placed robust setae, and a terminus with a fleshy, U-shaped pseudolabellum wrapped around an ellipsoidal mouth[6,19]. A

cibarial pump is present in Aneuretopsychidae, in addition to a second, smaller, labral pump probably homologous to the pharyngeal pump in the Dualulidae. In contrast to Mesopsychidae and Aneuretopsychidae, the much smaller, up to 2.2 mm long, forwardly projecting proboscis of Pseudopolycentropodidae bore on its outer surface obscure, regularly spaced, sclerotized annular rings with microtrichia and lacked pseudolabellae on the terminus[16,17,20]. There is evidence for a hypopharynx in compression Pseudopolycentropodidae, although its specific structure remains unknown. Other mouthpart elements, such as the labrum and labium, were used principally as proboscis braces. The maxillary palp consisted of three articles, the terminal one longer than the other two (Supplementary Note 2; Supplementary Fig. 7; Supplementary Data 4, 5). *Parapolycentropus* possessed a distinct cibarial pump under the clypeal region and a proboscis similar to other compression Pseudopolycentropodidae[16,17]. The proboscis bore setae and sclerotized bands encircling the outer surface. A distinct hypopharynx housed a bulbous pharyngeal pump connected distally to a salivary duct, although there is no evidence for a valve. The salivary duct had diminutive ventral serrations and terminated at the proboscis tip. The maxillary

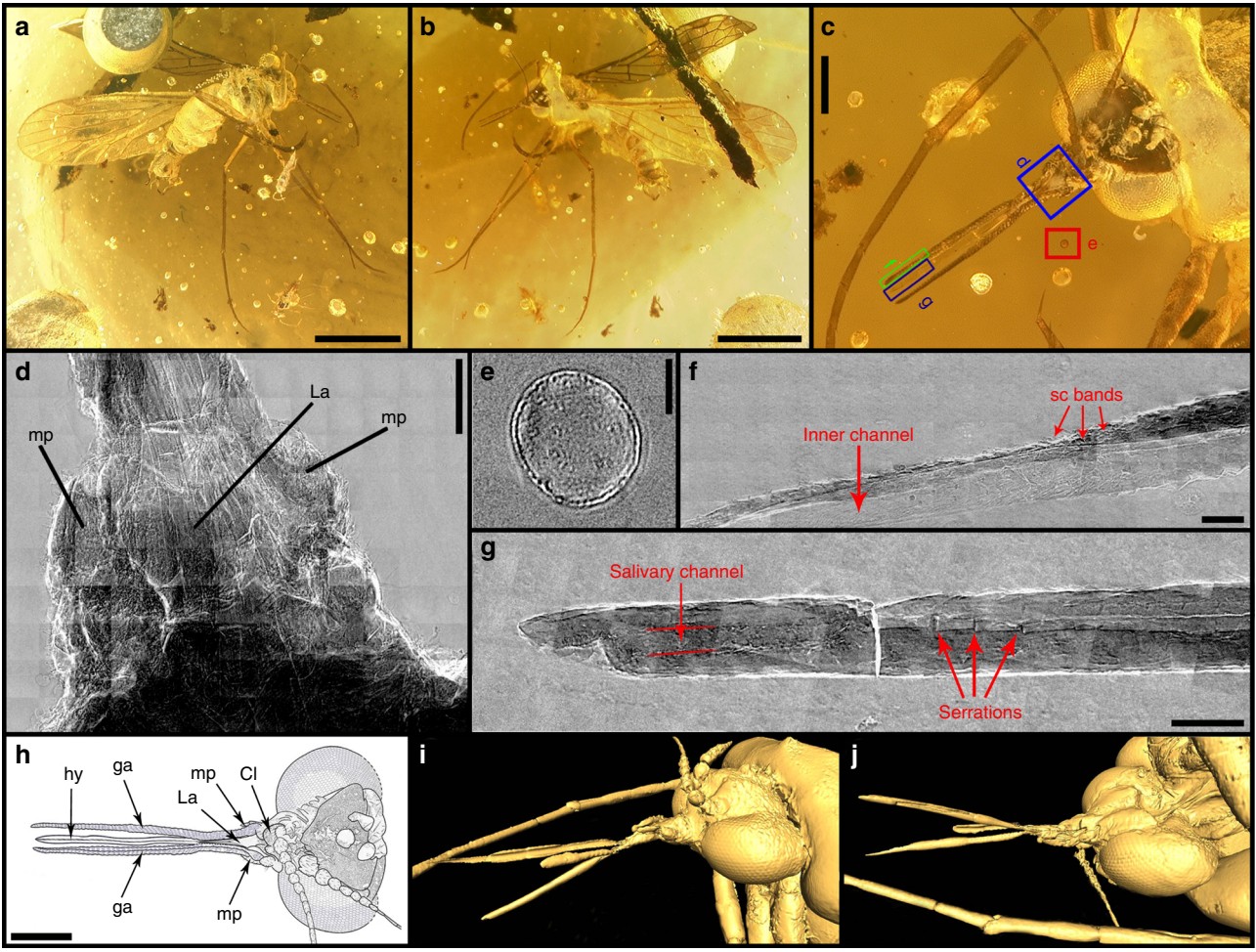

**Fig. 5** Nano-CT and Micro-CT images of *Parapolycentropus paraburmiticus*[17] head and associated pollen grain. Specimen CNU-MEC-MA-2015054; new material, male. **a** Insect in right lateral view. **b** Insect in left lateral view. **c** Head and mouthparts in dorsal view. **d** Nano-CT images of the proboscis base in ventral view, from the blue template in **c**. **e** Nano-CT images of a likely pollen grain near the right galea from the red template in **c**. **f** Nano-CT images of the left galeal tip from the green template in **c**. **g** Nano-CT images of the hypopharynx tip and associated external ornament in dorsal view, from the black template in **c**. **h** Line drawing of the head and proboscis in **c**. **i** 3-D reconstruction of head in lateral view from a micro-CT scan. **j** The same 3-D reconstruction of head in **i**, except in ventral view. Images **i** and **j** are from Micro-CT scanning, reconstructed in Amira software. Cl clypeus, ga galea, hy hypopharynx, La labrum, mp maxillary palp, sc bands sclerotized proboscis bands. Scale bars represent 1 mm in **a** and **b**; 0.2 mm in **c** and **h**; 30 μm in **d**; 10 μm in **e**; 15 μm in **f**; and 20 μm in **g**. Scale bars are absent in **i** and **j**

palps were similar to other Aneuretopsychina, short and consisting of three articles.

**Novel mouthparts of *Dualula*.** The feeding mechanism of *Dualula* consists of a pharyngeal pump linked by a valve to a salivary duct that provides controlled, outgoing salivary secretions. The pharyngeal pump–valve–salivary duct system is lodged within a food tube of much wider diameter. The food tube accessed incoming fluid food and was powered by a cibarial pump under the clypeus (Figs. 2 and 4; Supplementary Figs. 3a–e; 5a, b). This condition indicates a dual pump system that worked with fluids in the food tube and salivary duct secretions flowing in opposite directions. Recently, such a dual pump mechanism was considered possibly present in some long-proboscid Aneuropsychina[1,6], including Pseudopolycentropodidae based on compression-impression fossils from Northeastern China[6,20]. A dual pump system is better documented for *Parapolycentropus* (Fig. 5; Supplementary Figs. 8, 9). Similarly, the terminus of the *Dualula* proboscis was blunt, similar to a truncated straw end and lacked pseudolabellae or other terminal structures for sponging

surface fluids by capillary action[6,10,19] (Fig. 2g, i; Supplementary Fig. 5b). For a small, mosquito-sized insect such as *Dualula*, initial mobilization by enzyme-laden secretions of viscous surface fluids hidden in channels, funnels or other tubular structures of ovulate organs would have been an effective mode of ingesting pollination drops (Supplementary Note 3).

Four observations provide evidence for the function of the *Dualula* and closely related *Parapolycentropus* proboscis. The observations concern: (i) proboscis aspect ratios and diameters, (ii) proboscis cross sections, (iii) multiple longitudinal views of proboscis element surfaces, and (iv) expelled fluids from the food tube and salivary duct. First, *Dualula* possessed a proboscis considerably smaller and more slender than other long-proboscid scorpionflies of the mid Mesozoic. The average *Dualula* proboscis diameter was 0.12 mm, with a range of 0.10–0.16 mm. As a measure of slenderness, the average aspect ratio (proboscis length divided by its width) was 24.42 for a male and two female specimens. When compared to other long-proboscid, mid-Mesozoic scorpionflies such as Mesopsychidae with a much larger average body length (excluding antennae and proboscis) of 22.6 mm (Supplementary Data 4), the proboscis aspect ratio is

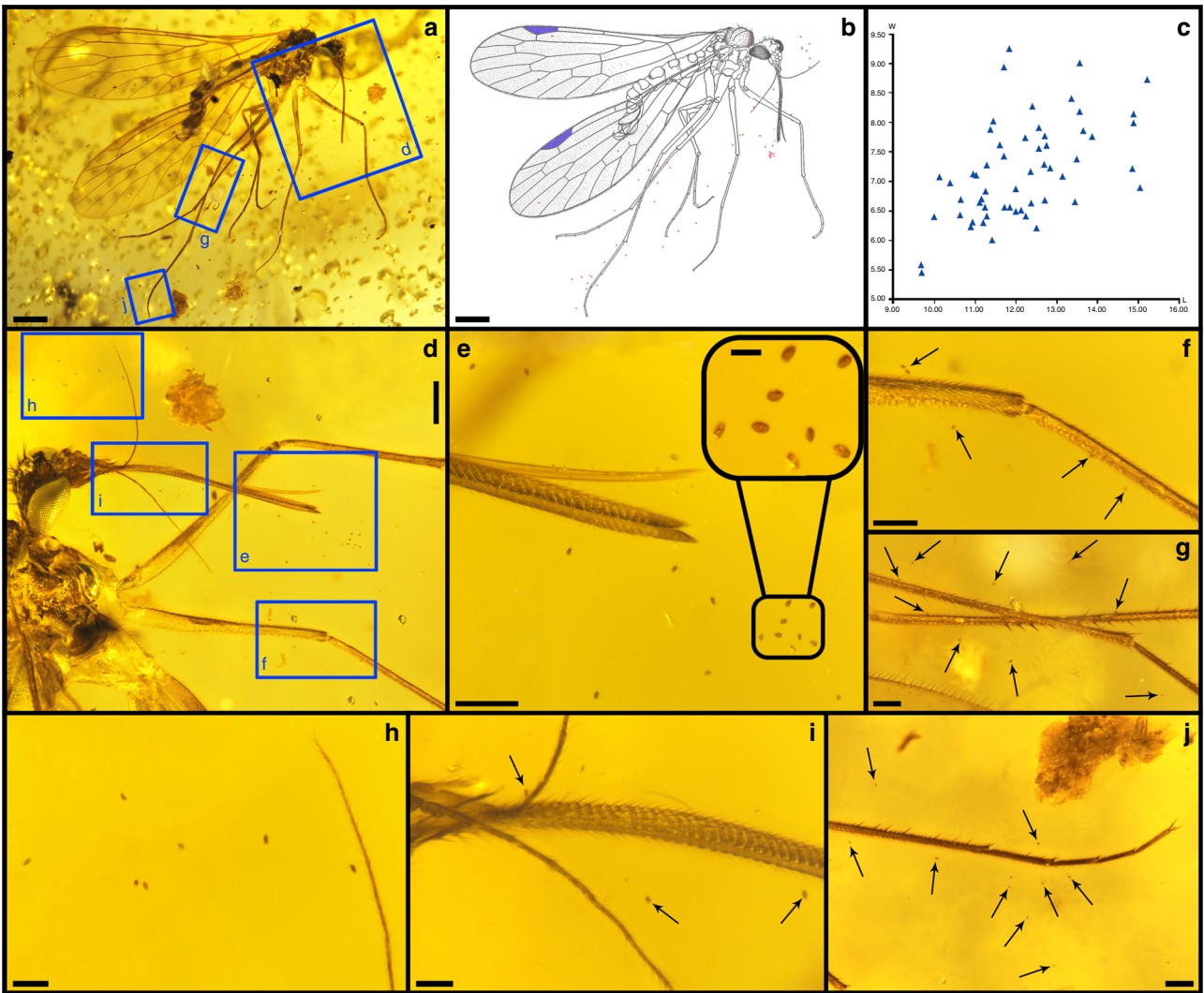

**Fig. 6** A male *Parapolycentropus paraburmiticus*[17] associated with Cycadopites sp. pollen grains. Concentrations of pollen surround the mouthparts, antennae, legs and wings (CNU-MEC-MA-2017012, new material). **a** The insect specimen. **b** Line drawing of the entire insect in **a** surrounded by *Cycadopites* pollen grains, shown as tiny red dots. **c** Plot of *Cycadopites* dimensions, shown as length (L) along the horizontal axis and corresponding width (W) along the vertical axis. The pollen-grain measurement data is from **a** and **b**, available in Supplementary Data 3. **d** Head, prothorax and proximal forelegs, enlarged from template in **a**. **e** Proboscis tip with galeae and hypopharynx surrounded by pollen grains, enlarged from **d**. Enlargement of several pollen grains near the proboscis tip at right. **f** Enlargement of right foreleg in **d**, with arrows pointing to nearby *Cycadopites* grains and clumps. **g** Enlargement of the right middle and hind legs in **a**, showing adjacent pollen indicated by arrows. **h** Pollen grains near the antennal tip outlined in **a**. **i** Proximal aspect of the proboscis and associated mouthparts with three pollen grains indicated by arrows. **j** Tarsus of right hind leg, indicted in **a**, with adjacent pollen indicated by arrows. Scale bars represent 0.5 mm in **a** and **b**; 0.1 mm in **e–g** and **j**; 0.2 mm in **d**; 50 μm in **h** and **i**; and 20 μm in-group of pollen grains from **e**

very similar at 22.46, but diameters are from three to four times that of *Dualula*. Comparisons of other lineages of Aneuretopsychina to *Dualula* indicate average proboscis aspect ratios are substantially lower and proboscis average diameters are considerably different. The average proboscis aspect ratio is 16.31 for Aneuretopsychidae; the average diameter is 0.35 mm (range 0.29–0.41 mm), about three times that of *Dualula*. For Pseudopolycentropodidae the aspect ratio is 13.02; the average diameter is 0.16 mm (range 0.08–0.25 mm), about 1.33 times wider than *Dualula*. For *Parapolycentropus* the aspect ratio is 14.03; the average diameter is 0.07 mm (range 0.04–0.11 mm), significantly narrower than *Dualula* (Supplementary Data 4 and 5). These variable proboscis aspect ratios and diameters strongly indicate accommodation to a variety of receiving diameters of tubular structures from a spectrum of contemporaneous gymnosperm

ovulate organs and possibly small angiosperm flowers[1,6] (Supplementary Fig. 11).

Second, a female *Dualula* proboscis is transversely cut where it intersects the amber surface (Supplementary Fig. 3e). This cross-section displays an expansive space between the inner surface of the galeal food tube and the outer surface of the encompassed salivary duct. The small salivary duct displays a very narrow, inner tube diameter. Third, longitudinal views of the outer surface of the galeal food tube and the salivary duct clearly is observed in several *Dualula* specimens (Fig. 2g, h; Supplementary Fig. 5a, b). The lengths of these salivary ducts are ca. 6–9 mm long and their inner diameters are about one third of their outer widths (Supplementary Fig. 3e).

Fourth, fluids are shown expelled from food tubes and salivary ducts soon after resin entombment, resulting in bubbles. Three

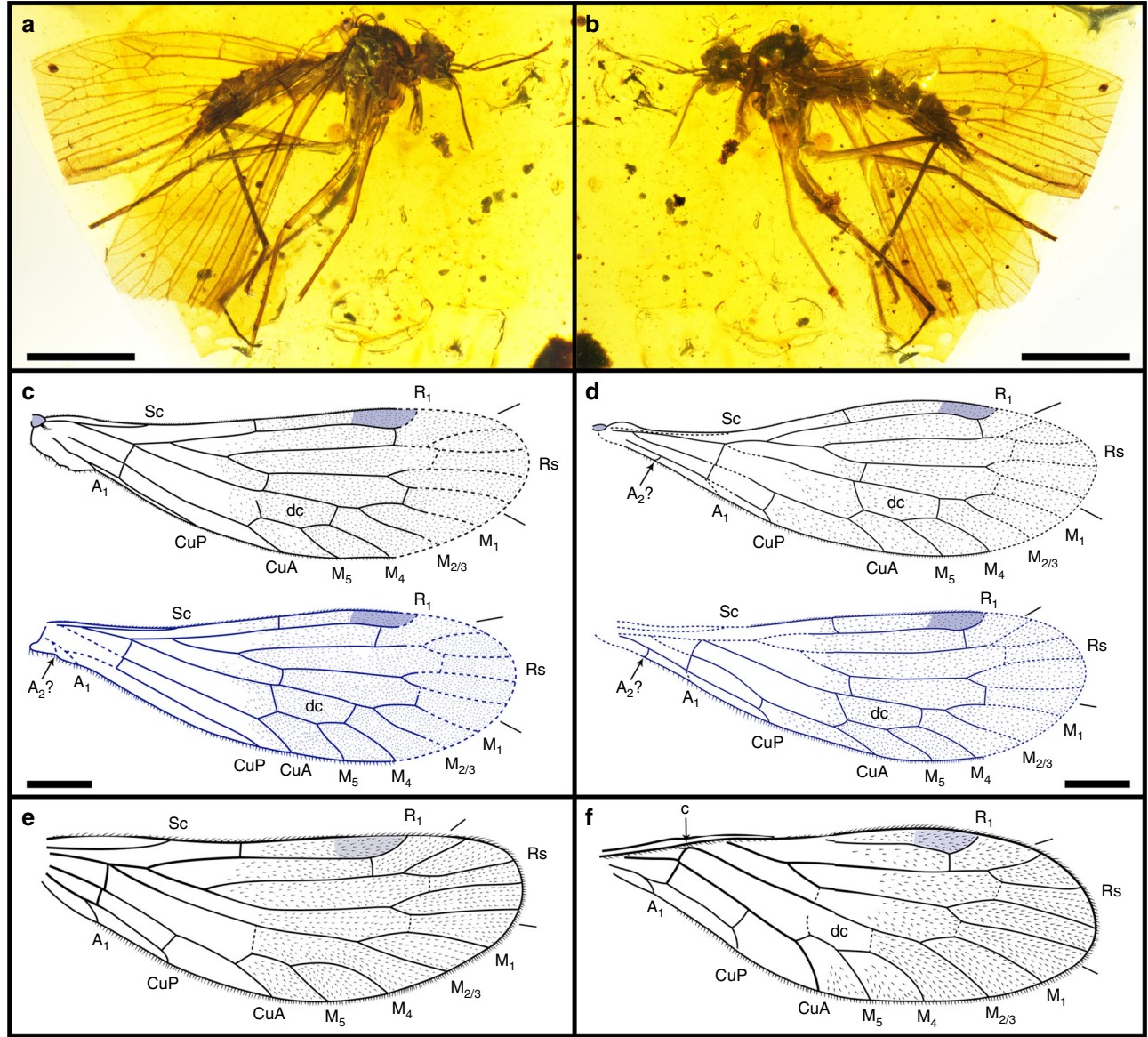

**Fig. 7** Photos and line drawings of a female four-winged *Parapolycentropus*. CNU-MEC-MA-2017006, new material with the redrawn images of forewings of two species of *Parapolycentropus*. **a** Specimen in right lateral view. **b** Same specimen at **a** in left lateral view. **c** Line drawings of right wings. **d** Line drawings of left wings. **e** Reconstructed forewing of *P. burmiticus*. **f** Reconstructed forewing of *P. paraburmiticus*. In **c** and **d**, forewings are black and hind wings are blue. Wings **e** and **f**, based on a published reconstruction in figure 8b and 8d of Grimaldi et al.[17]. Scale bars represent 1 mm in **a** and **b**, and 0.5 mm in **c**, **d** Subfigures **e** and **f** lack scale bars

examples show distinct fluid emissions in *Parapolycentropus* from the larger food tube (Supplementary Fig. 8p–r). In one example (Supplementary Fig. 8p), there is a bubble of food-tube fluid (top center) and a much smaller bubble of salivary-duct fluid (left center) from the same proboscis. These four types of evidence suggest not only presence of a double pump system, but also separate inflowing fluids into the food tube and outgoing secretions from a much narrower salivary duct (Supplementary Figs. 8i–m; 9d, e, g, h; Supplementary Movie 1).

**Feeding biology on host plants**. Two types of evidence are available for inferring the feeding biology of *Parapolycentropus* and *Dualula*. The first type is indirect evidence of structural features is consistent with proboscis probing and uptake of fluids from gymnosperm and angiosperm reproductive organs. Such evidence includes features of long-proboscis surfaces and suspect

gymnosperm and angiosperm reproductive organs that allow reception and accommodation of a proboscis. The second type is direct evidence demonstrating close association of pollen with body surfaces of insects[1,4,10,13,31,32]. Particularly important is identification of pollen to a source plant in the same deposit whose biology is consistent with insect pollination[4,13,32].

We examined 77 well-preserved specimens of *Dualula* and *Parapolycentropus*. Two specimens of *P. paraburmiticus* were associated with pollen grains adjacent their bodies. The first specimen had associated pollen grains of a gymnosperm (Fig. 6), and the other an unknown inaperturate grain (Fig. 5c, e). Pollen from the first specimen was distinctive and consisted of 54 smooth, monosulcate grains, olive-shaped in polar view, boat-shaped in longitudinal equatorial view and kidney-shaped in short equatorial view (Fig. 6e). The pollen is distinctly mono-sulcate, with the sulcus membrane thickening toward the margin, and unusually small, characterized by an average length of 12.15

µm (range 9.69–15.21 µm), an average width of 7.17 µm (range 5.46–9.24 µm) and an average length-to-width ratio of 1.70 (range 1.28–2.18 µm) (Supplementary Data 3). Based on these measurement and structural features, the pollen grains are attributed to *Cycadopites*[33], a gymnosperm form genus[34] (Supplementary Note 4). *Cycadopites* pollen is affiliated with Cycadales, Peltaspermales, Ginkgoales, Czekanowskiales, Pentoxylales and Bennettitales[34–37]. Some in situ *Cycadopites* from Bennettitales are quite small (down to 16 micrometers) and may be the source of the minute grains associated with *Parapolycentropus*[34]; however, other clades, especially Cycadales, cannot be ruled out. This occurrence provides direct evidence for a *P. paraburmiticus* feeding habit on pollination drops, indicating a pollinator relationship[38].

A single, possible pollen grain was found adjacent the proboscis of a *P. paraburmiticus* specimen (Fig. 5c, e). The surface details of this grain were distinct, based on a clear Nano-CT image (Fig. 5e). It is inaperturate, scabrate, nearly spherical, and 25.49 µm long by 22.15 µm wide. The affiliation of this grain is unclear. In addition, an examination of a large collection of Myanmar amber yielded 12 pieces with five well-preserved angiosperm flower morphotypes (Supplementary Fig. 11; Supplementary Data 6). The pieces contained one to a few flowers, and one included a branchlet of several clustered flowers (Supplementary Fig. 11k). Most of the flowers belong to *Tropidogyne* (Supplementary Fig. 11a, b, e–i, l–n, u), a possible member of the Cunoniacae (wild alder family), an extant family of early-derived, arborescent, dicotyledonous angiosperms of Oxalidales[40] with a Gondwanan distribution. *Tropidogyne* consists of two species – *Tropidogyne pikei*[39] and *T. pentaptera*[41]. Both flowers are cup-shaped, apetalous, bear five sepals, and house a ribbed, inferior ovary with a nectary disc and dark glands at the termini of floral appendages, features associated with insect pollination[31,42–44]. Members of Cunoniaceae produce tricolporate pollen, so clearly are not the producers of the grain associated with *P. paraburmiticus*. Two other cup-shaped flowers of unknown affinity are present (Supplementary Fig. 11c, d, j, k, v–z), designated Morphotypes A and B, that have features consistent with insect pollination[42–44] (Supplementary Note 4).

The longest, measured proboscis lengths for *Parapolycentropus paraburmiticus* is 1.53 mm and *P. burmiticus* 1.50 mm, which easily was accommodated, for example, by flower Morphotype A, a cup-shaped flower with an average sepal length of 2.44 mm but a likely corolla depth of about 1.55 mm (Supplementary Data 6). This floral distance from the top to the bottom of the corolla is in accord with the proboscis length of *P. paraburmiticus*, allowing for a reasonable 0.9 mm elevation of the gynoecium at the corolla base. Corolla depths of more bowl-shaped flowers of *T. pikei* and *T. pentaptera* were 1.89 mm and 1.97 mm, respectively, which would have accommodated proboscis lengths of both *Parapolycentropus* species. However, both *Tropidogyne* species would not have accommodated the much longer proboscis of *Dualula* that extended to 3.23 mm in one complete specimen (Supplementary Data 4). Modern, *Tropidogyne*-type flowers are consistent with an early-grade, basal angiosperm pollination mode typified by "small, bowl-shaped, white to yellowish, actinomorphic flowers, exposed sexual organs, perianth of separate sepals and petals [and] often clustered in inflorescences"[31]. Based on the indirect evidence of morphological features of *Tropidogyne* flowers and the direct evidence of gymnosperm pollen adjacent *Parapolycentropus* scorpionflies, this suggests that the pollinators of these two associations – one a gymnosperm host and the other several structurally similar angiosperm hosts – belonged to two taxonomically different insect pollinator guilds[1,4,31,43,44]. This suggests the plant host of *Dualula* was a gymnosperm ovulate organ[1,38] and the pollinator of *Tropidogyne* and similar flowers

may have been a small *Parapolycentropus*, but more likely syrphid and muscoid flies with sponging labellae[31,44]. These observations support: (i) pollinator activity between *Parapolycentropus* and gymnosperms; (ii) between *Tropidogyne* and related angiosperms and *Parapolycentropus*, syrphid flies and especially muscoid flies; and (iii) between *Dualula* and an unknown gymnosperm host[1,4,6,31,38,43,44].

**Hind wing evolutionary developmental biology**. Like proboscis uniqueness, vestigial hind wings also are a relevant feature that characterizes *Dualula* and *Parapolycentropus*. Extant and extinct lineages of scorpionflies normally bear two pairs of approximately equal sized and structurally similar membranous wings on the mesothorax and metathorax[5,22]. However, *Parapolycentropus* and *Dualula* from Myanmar amber (Supplementary Table 1; Supplementary Data 4 and 5) possess highly modified hind wings[16,17]. Furthermore, one genus of Liassophilidae (*Laurentiptera*)[45,46] and 11 described species of compression Pseudopolycentropodidae also bear hind wings reduced in size and venation. These related taxa often bear hind wings considerably smaller than their forewings and occasionally resemble halteres (Supplementary Fig. 12a, d); the hind wings of *Dualula* also have been reduced to small, haltere-like structures (Fig. 3c–e).

Of taxa with highly modified wings, one amber specimen possessed four wings of a nominal two-winged *Parapolycentropus* (Fig. 7a–d). The fore-and hind wings of this four-winged variant were identical in features to typical, four-winged scorpionflies past and present. The size, shape and venation of this specimen's wings are nearly identical between fore- and hind wings, minus minor differences of the anal area between right and left wings (Fig. 7c, d). No significant morphological or venational differences occurred between the forewings of the four-winged specimen and other specimens of this genus (Fig. 7e, f). The four-winged specimen demonstrates that hind-wing reduction is possible within a low-ranked, major lineage of Mecoptera (Fig. 7), an observation pertaining to Pseudopolycentropodidae, Liassophilidae, *Parapolycentropus* and *Dualula*. Given the protracted history of Mecoptera, the establishment of hind-wing reduction is a recurring evolutionary developmental pattern, explained by regulation of the *Ultrabithorax* Hox gene[47,48], transcription factors and regulatory cascades (Supplementary Note 5). The transition of Mecoptera hind wings from broad, membranous structures to small, narrow, haltere-like structures is a key structural acquisition that likely accelerated diversification of the group during the early Mesozoic, and continued as a ground-plan feature in earliest Diptera[49] (Supplementary Note 6).

**Male genitalia structure**. Male genitalia of one particular extant scorpionfly, Panorpidae (common scorpionflies), features a prominent structure arched over its abdomen resembling a large scorpion sting. Although scorpioid male genitalia is atypical for extinct and extant Mecoptera[50], mid-Mesozoic taxa of Aneuretopsychina, particularly Mesopsychidae[30,51], Pseudopolycentropodidae[52], *Parapolycentropus*[16] and *Dualula*, also exhibit unique, male genitalic homologies not found in other Mecoptera or Diptera. Distinctively homologous features of Pseudopolycentropodidae (Supplementary Fig. 12) and Mesopsychidae (Supplementary Figs. 13,14) from the Middle Jurassic include an upturned gonostylus, a very robust gonocoxa and dististylus (claspers), a terminal concavity on the dististylus, and undifferentiated cerci. These features were retained in most *Parapolycentropus* (Figs. 6a, b and 7a, b; Supplementary Figs. 9a, b; 15c,d) and *Dualula* (Supplementary Fig. 5c–e) from mid-Cretaceous Myanmar amber, showing a 65 million-year-long

evolutionary continuity of genitalic structure linking Eurasian Mesopsychidae, Pseudopolycentropodidae, *Parapolycentropus* and *Dualula*.

**Reproductive biology**. Fossil discoveries rarely provide insight into insect group behavior of the deep past. An exception is occasional pieces of amber that entomb a population of numerous, conspecific individuals engaged in congregation, such as mating or dispersal. Three Myanmar amber pieces preserve such swarming behavior and indicate a coordinated congregation of conspecific individuals that typically involve flies such as non-biting midges[53]. One mode of swarming behavior especially ubiquitous in many nematocerous fly lineages is lekking, a midair assembly of flies typically within a few meters of the ground surface, involved in a communal mating event. This phenomenon rarely has been documented in scorpionflies, but examples are known from modern Bittacidae[54] and fossil Nannochoristidae[55]. Because most Pseudopolycentropodidae, *Parapolycentropus* and *Dualula* species are mosquito sized and Myanmar amber pieces occasionally approach or exceed 5 cm in length, entombment of a swarm of lekking individuals is a distinct possibility (Supplementary Note 7). Fortunately, three pieces of amber were identified with abundant *Parapolycentropus* specimens, consisting of 9, 18, and 4 individuals, with varying combinations of *P. burmiticus* and *P. paraburmiticus* and female to male ratios of 1:6, 1:2, and 2:1, respectively. This pattern indicates an absence of species specificity and varied sex ratios, suggesting that lekking behavior favored male mating swarms[56]. Mating may have occurred in more diffuse combinations of *Parapolycentropus* species and irregular sex ratios involving aerial copulation of a larger female and more smaller males (Supplementary Fig. 16a, b) in an end-to-end stance of connecting genitalia (Supplementary Fig. 16c,d). Such an unusual copulatory position also exists among extant Panorpodidae and Panorpidae[50,57] (Supplementary Note 7).

## Discussion

Our reanalysis of Mecoptera employed a comprehensive list of 51 characters on 37 taxa that establishes a robust hypothesis for phylogenetic placement of core Pseudopolycentropodidae, *Parapolycentropus*, *Dualula*, other long-proboscid Aneuretopsychina, other mid-Mesozoic and modern Mecoptera, and early Mesozoic Diptera and Siphonaptera. Several morphological differences separate *Dualula* (Dualulidae) from other families of Mecoptera that include unique proboscis construction, reduced hind wings and genitalic features. The results (Fig. 1) indicate that Dualulidae is the sister group of *Parapolycentropus* but also has close relationships with other long-proboscid Mecoptera of Aneuretopsychina, basal Diptera and Siphonaptera lineages. Based on the results of trees in Fig. 1, there are two major hypotheses for the origin of the long-proboscid condition in mid-Mesozoic Mecoptera. The first hypothesis is the long-proboscid condition originated twice. Long-proboscid mouthparts were acquired in the common ancestor of the Pseudopolycentropodidae + (Liassophilidae + {Permotanyderidae + [Aneuretopsychidae + ||Mesopsychidae + Nedubroviidae||]}) clade and separately in the *Parapolycentropus* + Dualulidae clade, indicated by the magenta and brown arrows, respectively, in Fig. 1. A twofold origin would require that long-proboscid mouthparts were retained in Liassophilidae (*Liassophila*)[58] and Permotanyderidae (*Choristotanyderus*)[59], but originated independently in the *Parapolycentropus* + Dualulidae clade. Accordingly, they became generalized and present as haustellate mouthparts in the ancestor to Diptera and Siphonaptera.

The second hypothesis is the long-proboscid condition evolved three times. The first origination was the core

Pseudopolycentropodidae clade of *Pseudopolycentropodes* + (*Pseudopolycentropus* + *Sinopolycentropus*), indicated by the magenta arrow in Fig. 1. The second origination was the Aneuretopsychidae + (Mesopsychidae + Nedubroviidae) clade, indicated by the green arrow in Fig. 1. The third origination was the *Parapolycentropus* + Dualulidae clade, sister group to basal Diptera and Siphonaptera, indicated by the brown arrow in Fig. 1. This hypothesis presumes that Liassophilidae and Permotanyderidae retained the plesiomorphic condition of mandibulate mecopteran mouthparts. The threefold origin of long-proboscid mouthparts in Mecoptera is strongly favored here, because of distinct differences in proboscis construction among the three lineages[2,16,17,19,20,38,52].

Our phylogenetic result is similar to Ren et al.[6], but differs from other studies in four important aspects. First, basal Diptera are not the sister-group of Mecoptera, but rather originate within Mecoptera; Second, extant families of Mecoptera are not a monophyletic group, but exhibit paraphyly and polyphyly. Third, Aneuretopsychina are paraphyletic if Liassophilidae and Permotanyderidae lacked long proboscides. Fourth, *Parapolycentropus* is not a member of Pseudopolycentropodidae, but is a clade with *Dualula*, which in turn is the sister-group to basal Diptera + Siphonaptera. However, there are several limitations of our analysis. One issue is the lack adequate sampling, including all extinct and extant genera from families of Mecoptera and relevant Diptera. Second, the analysis is based on morphological data only. A third constraint was restriction of the data overwhelmingly to wing venation characters, which rendered insufficient resolution of Siphonaptera. To conclude, our research is a preliminary exploration of phylogenetic relationships among long-proboscid Mecoptera and relevant groups, and provides a framework for future studies. (These issues are discussed in Supplementary Note 1.)

The time of origin of the long-proboscid condition in Mecoptera, synonymous with the origin of the mid-Mesozoic clade Aneuretopsychina sensu lato[6,14], likely was late Permian. This timing is based on presence of Nedubroviidae[2], Mesopsychidae[3] and Liassophilidae[58] from late Permian to Middle Triassic deposits in Europe. However, Permotanyderidae, a likely member of the Aneuretopsychina, is known from the earlier late Permian of Australia[60]. Consequently, the earliest occurrences and likely place for the initial diversification of long-proboscid Aneuretopsychina was either northeastern Pangaea, on Baltica and Siberia, or northeastern Gondwana[61]. After this initial phase of modest speciation, a second phase of Aneuretopsychina diversification occurred during the Middle Jurassic to mid Cretaceous with multiplication of taxa in Mesopsychidae and Pseudopolycentropodidae, as well as speciation occurring in Aneuretopsychidae, *Parapolycentropus* and Dualulidae. This renewed diversification occurred in eastern Laurasia on the Tarim, Amuria, North China Block, South China Block and Annamia paleocontinents. These landmasses were docked earlier with eastern Laurasia or were separated by narrow oceanic gaps[61]. By mid Cretaceous, soon after Myanmar amber was deposited, the last lineages of the Aneuretopsychina became extinct, signaling the end of a 155 million-year legacy[1].

## Methods

**Localities and repositories**. This fossil study included 77 amber and seven compression fossil specimens. The amber specimens were collected from the Hukawng Valley of Kachin State, in northern Myanmar. The particular locality from which the specimens were collected was at the northern end of Noije Bum, which is a village located approximately at N26°150′, E96°340′, 18 km southwest of the town of Tanai. The amber is dated as earliest Upper Cretaceous (earliest Cenomanian), about $98.79 \pm 0.62$ Ma[18], equivalent to the early part of the Cenomanian Stage[62]. The compression fossils were collected from the latest Middle Jurassic Jiulongshan Formation at Daohugou Village, Shantou Township, Ningcheng County of Inner Mongolia, China. This locality is located at N41°18.979′,

E119°14.318′, and has been radioisotopically dated at 164 Ma[63], corresponding to the later part of the Callovian Stage[62]. Most of the studied material is housed in the Key Lab of Insect Evolution and Environmental Changes, at the College of Life Sciences of Capital Normal University (CNU), in Beijing, China. Six specimens of CNU-MEC-MA-2015025, CNU-MEC-MA-2015027, CNU-MEC-MA-2015029, CNU-MEC-MA-2015030, CNU-MEC-MA-2015031 and CNU-MEC-MA-2015032 currently are on loan to CNU but will be returned to the Three Gorges Entomological Museum (EMTG), in Chongqing, China, where they finally will reside.

**Amber preparation.** All amber pieces were polished with emery paper sheets with varying grit sizes of 300, 600, 1000, 3000, 5000 and 7000 grit, in a time sequence of coarse to finer grit size. Care was taken to avoid contamination from sheets of different grit size. The amber finally was processed with Tamiya polishing compound@2004 TAMIYA for a smooth finish. For *Parapolycentropus paraburmiticus* (CNU-MEC-MA-2017012), the area of interest was polished close to the insect body surface for ease of viewing, while avoiding contact damage to body structures. However, the isolation of pollen grains was not feasible. Imaging of the *Cycadopites* sp. pollen grains by a Micro-CT scanner also proved unsuccessful, attributable to the poor absorptive capacity of X-rays from the lack of a density difference between the pollen and entombing amber.

**Specimen imaging.** Most of specimens were examined and photographed under a Nikon SMZ25 microscope attached to a Nikon DS-Ri2 digital camera system in the Fossil Insect Laboratory at CNU. Four specimens – CNU-MEC-MA-2016007, CNU-MEC-MA-2015030, CNU-MEC-MA-2015038 and CNU-PLA-MA-2016001 – were photographed under an Olympus DSX100 digital camera system. The equipment was the Scanning Electron Microscope (SEM) Laboratory of the National Museum of Natural History (NMNH), Smithsonian Institution, Washington, D.C. Photographs of other specimens, such as CNU-MEC-MA-2015054, were captured with a Z16 Leica®TM lens attached to a JVC KY-F75U digital camera system in the Department of Entomology Laboratory at NMNH[64–66]. This system was used to stack photos employing a series of software consisting of Cartograph 7.2.5®TM and Archimed®TM 6.1.4, and stacked with Combine ZP®TM. Incident lighting was used by techniques suggested in summaries of best procedures[64–66]. All photomicrographs with green background (Supplementary Fig. 5b) were taken by green epifluorescence as the light source, attached to a Zeiss Axio Zoom.V16 compound microscope, and with a fluorescence-image noise elimination system (Zeiss Apo Tome.2) in the College of Life Sciences public laboratory at CNU. Micro-CT scanning and three-dimensional reconstruction of specimens CNU-MEC-MA-2015054 and CNU-MEC-MA-2017008 were scanned with a Micro-CT (Nano Voxel 3000D, Sanying Precision Instruments Co., Ltd., Tianjin, China), located at the School of Mathematical Sciences at CNU. The voltage of the Micro-CT scanner was 50KV and the phase-contrast enhancement technique was used to reconstruct CT images with a higher contrast. The proboscis structures of the two above specimens were rendered with Amira@ 5.4.3 (Visage Imaging, San Diego, USA) and Avizo@ Fire 8.0 (Visualization Sciences Group; Massachusetts, USA). The Nano-CT images of specimen CNU-MEC-MA-2015054 – including the insect and pollen grain – were scanned with a Nano-CT (BL01B1) located in the National Synchrotron Radiation Research Center (NSRRC), in Hsinchu, Taiwan. Three SEM photos of specimens CNU-MEC-NN-2016001P, CNU-MEC-NN-2016008 and CNU-MEC-NN-2016015P were completed in the SEM Lab of the NMNH, under the PHILIPS XL 30 ESEM system. The figures were composited using Adobe Photoshop CC graphics software, and the line drawings were prepared by Adobe Illustrator CC and Adobe Photoshop CC graphics software.

**Measurements, abbreviations and terminology.** The lengths of the proboscides, wings and antennae were measured from the base to apex. The body lengths were measured from the apex of the head to the appendicular terminalia of the abdomen, excluding the antennae and proboscis. The widths of the proboscides were measured at their broadest dimension, excluding the labrum and maxillary palpus. The lengths of pollen grains were measured through the horizontal axis and widths were measured by the vertical axis approximately perpendicular to the horizontal axis.

The terminology of wing venation for Pseudopolycentropodidae, *Parapolycentropus* and Dualulidae follows established nomenclature[16,17]. Corresponding abbreviations in the text and figures are the following. For wing venation: Sc subcosta, $R_1$ first branch of the radius, Rs radial sector, $M_1$ first branch of the media, $M_2$, second branch of the media, $M_3$ third branch of the media, $M_4$ fourth branch of the media, $M_5$ fifth branch of the media, $M_{2/3}$ second and third branches of the media, MA anterior media, MP posterior media, CuA anterior cubitus, CuP posterior cubitus, $A_1$/1A first branch of the anal vein, $A_2$/2A second branch of the anal vein, $A_3$ third branch of the anal vein, and dc central discal cell. For head and proboscis: Ant antennae, car cardo, CE compound eye, Cl clypeus, fc food canal, ga galea, hy hypopharynx, is inner surface of galea, La labrum, mp maxillary palp, oc ocellus, os outside surface of galea, Pr proboscis, sc sclerotized bands, sti stipes. For genitalia: c cercus, epi epiphallus, go bas gonocoxa basistylus, go dis gonocoxa dististylus, par paraprocts, pm paramere, p penis, pe penunci, spa superanale, sV-sIX fifth to ninth sterna, and tVI-tIX, sixth to ninth terga.

**Nomenclatural acts.** This published work and the nomenclatural acts it contains have been registered in ZooBank, the proposed online registration system for the International Code of Zoological Nomenclature (ICZN). The ZooBank LSIDs (Life Science Identifiers) can be resolved and the associated information viewed through any standard web browser by appending the LSID to the prefix "http://zoobank.org/". The LSIDs for this publication are urn:lsid:zoobank.org:pub:8E7D07F9-A618-48D6-8EEC-F5AC68593C5C (for publication); urn:lsid:zoobank.org:act:A219BB2D-209F-4D2E-ABAA-AB10CB8CF0D8 (for Dualulidae fam. nov.); urn:lsid:zoobank.org:act:E9F85E03-B6C6-41FF-A82D-16EE9CFC21C0 (for *Dualula* gen. nov.); urn:lsid:zoobank.org:act:70C21743-5FEF-48C1-8FE4-7739BC029394 (for *Dualula kachinensis* sp. nov.).

**Reporting summary.** Further information on experimental design is available in the Nature Research Reporting Summary linked to this article.

## Data availability
The authors declare that the data supporting the findings of this study are available within the paper and its Supplementary Information Files. Higher-resolution versions of the figures (https://doi.org/10.6084/m9.figshare.7775801.v1) and supplementary data (https://doi.org/10.6084/m9.figshare.7775822.v1) have been deposited in the figshare database. All relevant data are available from the corresponding authors upon request.

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

## Acknowledgements

We thank Matthew L. Buffington and Elijah J. Talamas of the Systematic Entomology Laboratory in Washington, D.C. for assistance in using the Z16 Leica microscope and image processing software in the NMNH Entomology Lab. We are grateful to Scott Whittaker, SEM lab manager, for guidance in specimen preparation and use of the Philips environmental scanning electron microscope system. Junjie Wang assisted in acquiring images of nano-CT and micro-CT scanners at the National Synchrotron Radiation Research Center in Taiwan, and Shiwo Deng provided help in use of Micro-CT instruments at Mathematical Sciences in Capital Normal University (CNU). Taiping Gao, Yongjie Wang and Longfeng Li of CNU provided valuable comments and suggestions. We acknowledge the online Paleobiology Data Base for accessing fossil record data. This research is supported by grants from the National Natural Science Foundation of China (grants 31730087, 41688103 and 31672323), Program for Changjiang Scholars and Innovative Research Team in University (IRT-17R75), and Support Project of High-level Teachers in Beijing Municipal Universities (IDHT20180518). X.D.L. is supported by the Graduate Student Program for International Exchange and Joint Supervision at Capital Normal University (028175534000). The research of Carol Hotton was supported in part by the Intramural Research Program of the National Institutes of Health, National Library of Medicine of the United States. This is contribution 367 of the Evolution of Terrestrial Ecosystems consortium at the National Museum of Natural History, Smithsonian Institution, in Washington, D.C.

## Author contributions

D.R. and C.C.L. designed the experiments. X.D.L., C.K.S., and D.R. contributed materials and analytical tools. X.D.L. took the photographs. X.D.L. and C.C.L. made the line drawings and reconstruction. X.D.L., C.C.L., and C.L.H. were responsible for palynology and entomophily inferences. X.D.L., C.C.L., C.K.S., C.L.H., and D.R. performed the analyses, experiments and wrote the manuscript. All authors read and approved the final manuscript.

## Additional information

**Competing interests:** The authors declare no competing interests.

