## [Peer Review File · Nature Communications]

Reviewers' Comments:

Reviewer #1:

Remarks to the Author:

Excellent manuscript

Congratulations

Antonio Arillo

Reviewer #2:

Remarks to the Author:

The manuscript is an important contribution to systematic entomological palaeontology, with compelling and detailed account of the feeding structures of new long-proboscid Mecoptera from Burmese amber. My major concern with the paper is the limited phylogenetic study that proposes the order Diptera arose from these groups.

The phylogenetic analysis is limited in a number of ways:

1. The Diptera belong to the Antliophora, a group of 3 orders, Siphonaptera, Mecoptera and Diptera. The authors exclude Siphonaptera from their analysis because the lack wings. This is not an appropriate reason to exclude them, and the Siphonaptera should be included in the analysis.
2. The matrix of the analysis is dominated by wing venation characters visible in fossils. Something like 60% of the characters in the 51 character analysis are from wings alone, presumably because these can be scored from fossils. However the matrix also includes internal characters of the proventriculus that cannot be scored in fossils. This dominance of wing characters biases the results of the analysis. In extant groups the details of wing venation (such as scored in this ms) are known to be subject to high levels of homoplasy.

Recent molecular analyses have shown that in the Antliophora, Diptera is sister to (Mecoptera+Siphonaptera). This has been shown in Sanger molecular analyses Wiegmann et al (2009, BMC Biology), and large phylogenomic analyses (supplementary Figure 1c in this ms, from almost 1,500 genes), yet their comparison of studies in Supplementary figure 1 poses this result as just 1 among a number of studies with far fewer characters and limited character sampling.

In short, the authors need to deal more comprehensively in their study of the incongruence in their study between their results and previous results. No recent analysis has posed that the entire Mecopteran crown group is rendered paraphyletic by Diptera. In some cases Mecoptera are rendered paraphyletic by Siphonaptera, however the authors could not test this result because they excluded Siphonaptera from their analysis.

In addition, the phylogenetic tree reported in the ms suffers from very low support in a number of critical nodes. For example, the green-arrowed node in Figure 1 of the ms is supported by a single character, #40, and has an almost zero bootstrap value of 7. This character refers to the width of the anal area of the forewing at its base. This paper of the wing is extremely variable in extant and fossil groups, and often is correlated with size-the smaller the insect, the more narrow to anal area of the wing.

In summary the paper is not up to the standard required of Nature Communications. There is some

beautiful systematic paleontology in the ms. The phylogenetic analysis is flawed, and the authors do not address the glaring differences between their phylogenetic results and recent phylogenetic and phylogenomic analyses. i recommend rejection.

Reviewer #1 E-mail Comment

Comment 1. “Excellent manuscript
Congratulations
Antonio Arillo”

Authors’ Response. Thank you for your appreciation of our work. We have done considerable amount of work in documenting the morphology and habits of this fantastic group of medium size to miniature insects. In addition, the phylogenetic analysis shows the classification position of these long-proboscid taxa, and reveals their relationships with other Mecoptera, basal Diptera and Siphonaptera. There are four other major contributions. First, we found a new type of siphonate mouthparts, detailed its morphological structures and compared these mouthparts to other documented groups. Second, the first record of direct evidence for pollination is presented for long-proboscid scorpionflies. The pollen grains associated with the scorpionfly body is attributed to an extinct gymnosperm lineage, and a single pollen grain possibly may be an early angiosperms. Third, we have provided a new evolutionary developmental perspective for understanding the change from a four-winged to a two-winged mecopteran. This hypothesized transformation is based on how one Hox-controlled gene, *Ultrabithorax (Ubx)*, could implement such a transformation from membranous wings to haltere-like structures. Fourth, based on several amber specimens detailing the swarming and mating of certain scorpionfly species, we have the first, direct behavioral evidence of scorpionfly reproductive biology. These two-winged groups very likely engaged in a mating flight behavior, probably as part of a swarm, and end-to-end (tip-to-tip) copulation with female genitalia in a reversed position. Furthermore, we believe that this study of long-proboscid two-winged scorpionflies may spur additional future investigations into the phylogenetic relationships between Mecoptera and relevant derivative clades, such as Diptera and Siphonaptera.

Reviewer #2 E-mail Comments

Comment 1. “The manuscript is an important contribution to systematic entomological palaeontology, with compelling and detailed account of the feeding structures of new long-proboscid Mecoptera from Burmese amber. My major concern with the paper is the limited phylogenetic study that proposes the order Diptera arose from these groups.”

Authors’ Response. The main purpose of our phylogenetic analysis is to clarify the taxonomic position of new family, Dualulidae, and to explore the taxonomic relationship among these two-winged scorpionflies and other long-proboscid groups, rather than to completely elucidate and resolve the systematic classification of the Antliophora. Excluding the phylogeny,

our study also has the following four highlights, as alluded to by Reviewer 2. First, we described details of a new type of mouthparts, which has a special double-pump structure for feeding on surface fluids. Second, we report direct evidence for the long-proboscid mecopteran pollination, with pollen grains attached to the insect body. These pollen grains support the symbiotic relationship between gymnosperms and early angiosperms. Third, from the perspective of evolutionary and developmental biology, we explore the possible causes of hind wing vestigiality in Mecoptera. Fourth, based on the direct evidence of amber fossils, our understanding has been augmented of the reproductive biology of these extinct scorpionfly lineages. Although we consider these discoveries to be important, the phylogenetic analysis and resulting classification also is an important, albeit subsidiary, part of the manuscript, but is only a preliminary exploration among phylogenetic relationships among the long-proboscid Mecoptera and relevant groups, extinct and extant, with an intention to provide a framework for future studies.

Comment 2. “The phylogenetic analysis is limited in a number of ways:

1. The Diptera belong to the Antliophora, a group of 3 orders, Siphonaptera, Mecoptera and Diptera. The authors exclude Siphonaptera from their analysis because the lack wings. This is not an appropriate reason to exclude them, and the Siphonaptera should be included in the analysis.”

Authors’ Response. We agree with Reviewer 2. We added Siphonaptera in phylogenetic analysis and the new character-state matrix has been updated. We used the same phylogenetic procedure as before, but with updated data. The new resulting trees are similar to the original analysis that did not include Siphonaptera. The new trees show that Mecoptera are indeed paraphyletic. In addition, basal Diptera and Siphonaptera form a branch that has a sister-group relationship with *Parapolycentropus* and Dualulidae. It is important to note that only nine families of modern scorpionflies hitherto (six families are represented in the data matrix), which is only 23 % of all 39 mecopteran groups. Additionally, because of the vagaries of fossil preservation, most fossil insect specimens do not have well-preserved body structures, especially many early Mecoptera and basal Diptera from compression material. (On the other hand, these imperfect fossils represent valuable character-state data that could never be retrieved from the modern antliophoran fauna.) Because of all these factors, our analysis can only be treated as a preliminary assessment, but not fulfill complete resolution of the phylogenetic relationships of groups among the Antliophora. Nevertheless, it is essential in a study of this type to elucidate the phylogenetic relationships and position in antliophoran classification of the new family Dualulidae. In this process, we have documented several morphological similarities to indicate that *Parapolycentropus* and Dualulidae are similar to early Diptera. The view that

Pseudopolycentropodidae, as a group within Aneuretopsychina, are similar to Diptera was already proposed in Novokshonov (1997) and other relevant studies:

Novokshonov, V. G. Some Mesozoic scorpionflies (Insecta: Panorpida = Mecoptera) of the families Mesopsychidae, Pseudopolycentropodidae, Bittacidae, and Permochoristidae. *Paleontol. J.* 31, 65–71 (1997).

Grimaldi, D. A., Zhang, J., Fraser, N. C. & Rasnitsyn, A. P. Revision of the bizarre Mesozoic scorpionflies in the Pseudopolycentropodidae (Mecopteroidea). *Ins. Syst. Evol.* 36, 443–458 (2005).

Grimaldi, D. A. & Johnston, M. A. The long-tongued Cretaceous scorpionfly *Parapolycentropus* Grimaldi and Rasnitsyn (Mecoptera: Pseudopolycentropodidae): new data and interpretations. *Am. Mus. Novit.* 3793, 1–24 (2014).

Comment 3. “The matrix of the analysis is dominated by wing venation characters visible in fossils. Something like 60% of the characters in the 51 character analysis are from wings alone, presumably because these can be scored from fossils. However the matrix also includes internal characters of the proventriculus that cannot be scored in fossils. This dominance of wing characters biases the results of the analysis. In extant groups the details of wing venation (such as scored in this ms) are known to be subject to high levels of homoplasy.”

Authors’ Response. As mentioned previously, due to the taphonomic vicissitudes of fossil preservation, delicate and soft-bodied insects for a majority of fossil insect specimens – specifically compression fossils – do not have well-preserved body structures for detailed morphological studies. However, this is not the case for amber specimens, which retain very delicate structures comparable with that of modern insect material which are the source for most of the focal insect lineages in our study. Consequently, the fossil record is mixed; half of it is less than ideal, particularly compression material before the Cretaceous Period; but the other is superb, comparable to that of modern material for amber specimens after the Jurassic Period. Although many specimens have only records of wings, and some may have preserved only a single fore- or hind wing, the majority of extinct insect families are classified according to the features of wings, such as wing shape, wing venation, nygmata, setation and wing-fringe elements. However, about 60% of the characters used in our phylogenetic analysis refer to wing shape, venation and other wing features, a relatively low percentage of wing characters compared to other such studies.

There have been three published phylogenetic analyses focusing on mid-Mesozoic Mecoptera phylogeny. The first analysis was in Ren *et al.* (2009). This phylogenetic analysis was based on 53 morphological characters of most extinct and extant families of Mecoptera, Siphonaptera and Diptera, of which 28 (53 %) were wing-related features. A second analysis was published by Lin *et al.* (2016), which were a phylogenetic analysis to clarify the genus-level relationships within Mesopsychidae, a family of Aneuretopsychina, and explore the origin of the siphonate proboscis. The character-state matrix included 26 morphological characters, only two of which were non-

wing body features. The third analysis was Wang *et al.* (2012), which was a preliminary cladistic analysis of Cimbrophlebiidae consisting of eight taxa and seven morphological characters, all associated with wings. Our reading of the literature indicates that wing venation is relatively conservative and stable in the Mecoptera, especially in extinct groups. (This may or may not be the case in other major fossil insect lineages.) While the left and right wings are not completely symmetrical in few groups, among which the principal veins there are consistency, and the differences that we do observe refer only to infrequent vein sub-branches of individual primary vein branches, or occasional presence of additional cross-veins. Consequently, we assess our data set as valid based on three criteria. First, there are a relatively low proportion of vein characters (60 %) in our data compared to other such studies of the Mecoptera. Second, there is relative stability of the vein system in fossil Mecoptera. Third, there is recognition of rare and occasional, anomalous deviations of vein position and branching when it does occur in fossil scorpionflies.

Comment 4. “Recent molecular analyses have shown that in the Antliophora, Diptera is sister to (Mecoptera+Siphonaptera). This has been shown in Sanger molecular analyses Wiegmann *et al.* (2009, BMC Biology), and large phylogenomic analyses (supplementary Figure 1c in this ms, from almost 1,500 genes), yet their comparison of studies in Supplementary figure 1 poses this result as just 1 among a number of studies with far fewer characters and limited character sampling.”

Authors’ Response. We appreciate Reviewer 2 for reminding us this important reference, and we have added this source as part A of Supplementary Note 1. As the purpose of this section is to review the history of phylogenetic analysis within Holometabola, we listed all relevant results, and should not have ignored those studies with sampling limitations or inclusion of a limited number of characters. Indeed, none of these conclusions can represent the authentic evolutionary history of Holometabola. Neither the analyses of Wiegmann *et al.* (2009) nor Misof *et al.* (2014) considered the role of fossil records in the evolution of Antliophora, so their conclusions only show the relationships among extant groups. Some analyses employ both molecular and morphological data, so called total evidence, such as Whiting *et al.* (1997), Trautwein *et al.* (2012) and Peters *et al.* (2014). However, currently such studies have not included (all) fossil and modern groups. Consequently, our assessment provides a new approach for future analysis and reminds researchers of the importance of extinct groups in evolutionary and phylogenetic studies.

Comment 5. “In short, the authors need to deal more comprehensively in their study of the incongruence in their study between their results and previous results. No recent analysis has posed that the entire Mecopteran crown group is rendered paraphyletic by Diptera. In some cases Mecoptera are rendered paraphyletic by Siphonaptera; however, the authors could not test this result because they excluded Siphonaptera from their analysis.”

Authors' Response. We agree. We added a paragraph in the Discussion Section to compare our results with other published phylogenetic analyses. We acknowledged clearly that there are limitations in our analysis, and this result is only a preliminary exploration that the phylogenetic relationships among Mecoptera and relevant groups of Diptera and Siphonaptera, which cannot be solved completely at this time. However, our results do show where more intensive fossil collection activity should be focused, particularly on lineages of the Aneuretopsychina, Liassophilidae Permotanyderidae, Dualulidae, *Parapolycentropus*, and the earliest lineages of Diptera and Siphonaptera. This concentration of research activity would not have been available if previous studies were the guide.

As we now include Siphonaptera in the new analysis, the results indicate that Mecoptera, Diptera and Siphonaptera are paraphyletic groups. It has not been the first time that this conclusion has been arrived at; for example, see Fig. 3 in Ren *et al.* (2009), where Diptera, Liassophilidae and Permotanyderidae form a clade, and Siphonaptera lies within Mecoptera. The current analysis supersedes the previous analysis that lacked inclusion of these two-winged, mostly amber-preserved groups. Our analysis is complemented the systematic classification of these unique groups, and demonstrated that they have close relationships with basal Diptera.

Indeed, numerous analyses based on molecular data show that the Mecoptera has a distant relationship with Diptera, but a close one with Siphonaptera. However, the molecular data can only be applied to modern groups, and not to more numerous, major extinct groups, of which Mecoptera is a prime example. In other words, it is unreasonable to use only a small number of nine modern families to represent the entire order of 39 families. In a sense, there is a tradeoff between a greater geochronological coverage (274 million years) and less than ideal overall character-state resolution, versus no geochronological coverage (the geological instant of today) with much better character-state resolution. We would augur that both approaches need to be taken. Therefore, the result cited by Reviewer 2 only reflects the relationship between Mecoptera, Diptera and Siphonaptera in modern groups (the latter option), but they cannot represent fossil groups (the former option). Our research addressed this shortcoming and may open up a new door for the future research.

Comment 6. "In addition, the phylogenetic tree reported in the ms suffers from very low support in a number of critical nodes. For example, the green-arrowed node in Figure 1 of the ms is supported by a single character, #40, and has an almost zero bootstrap value of 7. This character refers to the width of the anal area of the forewing at its base. This paper of the wing is extremely variable in extant and fossil groups, and often is correlated with size-the smaller the insect, the more narrow to anal area of the wing."

Authors' Response. We agree with Reviewer 2. We have deleted the relevant issue, indicated as a green arrow in Fig. 1 in the discussion of the main text. As the body structures are poorly preserved in fossil-compression groups, most of the features are almost of necessity wing-related. At the same time, most extinct groups are distinguished by the wing venation, frequently lacking other body features. The evolutionary changes of wing venation are generally faster than other parts of the insect body, with parallelisms and reversals present. Consequently, the support for individual branches is relatively low. This result often is inevitable in studies of extinct groups, and can be changed only if better-preserved fossils are discovered in the future. However, the fundamental evolutionary direction (polarity) of wing venation still can be ascertained; for example, wing venation from complex to simple, the number of branches from more to less and the shift from separate veins to fused veins. Our research still presents significance in elucidation of relationships among the long-proboscid groups and other Mecoptera.

Comment 7. "In summary the paper is not up to the standard required of *Nature Communications*. There is some beautiful systematic paleoentomology in the ms. The phylogenetic analysis is flawed, and the authors do not address the glaring differences between their phylogenetic results and recent phylogenetic and phylogenomic analyses. I recommend rejection."

Authors' Response. We admit that there are some limitations in our phylogenetic analysis, which we have addressed. These issues, given the nature of the fossil record, cannot completely resolve the phylogenetic relationships of Mecoptera and relevant groups, but they can be instructive and suggest where additional work needs to be focused. Our phylogenetic analysis also secures the phylogenetic position of the Dualulidae, which was the major reason for the analysis. We also understand that analyses of modern groups only, with molecular and morphological data, cannot address the history of the Mecoptera and its derivative major lineages because of the absence of a vast number of extinct lineages that cannot be taken into account in analyses based on modern-only and molecular data. (Parenthetically, we state that this issue is even a more crucial problem in phylogenetic relationships of angiosperms to other Mesozoic seed plants, where it is unclear which of several seed-plant groups angiosperms are related to. At least in the Antliophora, based on character-state and lineage data, there is a much better sense of relationships.) We have addressed the concerns of Reviewer 2 by re-evaluating our character-state matrix and inclusion of the Siphonaptera. Importantly, our major significance involves clarifying the taxonomic position of new family, Dualulidae, and its phylogenetic relationships, paraphyletic or otherwise, to other Mecoptera, early Diptera and Siphonaptera. We have added

this updated context and explanation in Discussion Section of main text and Supplementary Note 1.

We hope these corrections and revisions will be more than satisfactory for potential acceptance in *Nature Communications*.

Sincerely yours, and on behalf of our coauthors,

Dr. Conrad Labandeira
Senior Research Scientist
and Curator of Fossil Arthropods

Prof. & Dr. Dong Ren
College of Life Sciences
Capital Normal University
Beijing, 100048, China

Reviewers' Comments:

Reviewer #2:

Remarks to the Author:

The authors have made some changes that I suggested and made clear the limitations of their phylogenetic analysis.

Reviewer #2 Comment

Comment 1. “The authors have made some changes that I suggested and made clear the limitations of their phylogenetic analysis.”

Authors' Response. We appreciate the critical eye of Reviewer 2 regarding our original phylogenetic analysis. We previously were aware of the limitations in our phylogenetic analysis, and now have added this explanation in the main text and supplementary information. Interestingly, inclusion of Siphonaptera in the new analysis, based on Reviewer #2's suggestion, did not alter appreciably the topology of the new, resulting trees. The new trees of the second analysis are quite similar to the original ones. Nevertheless, exploration of the origin of the longproboscoid condition based on the current cladistic analysis did improve our understanding of the evolution of the haustellum in early Mecoptera, Diptera and Siphonaptera. We hope this research can provide enlightenment and baseline for the future work.

We hope these modifications meet the requirements of Nature Communications, and the manuscripts can be published in a timely fashion. Please do not hesitate to ask for additional information from the corresponding authors.